# Understanding the role of veterinarians in antimicrobial stewardship on Canadian dairy farms: A mixed-methods study

Claudia Cobo-Angel[1,2,‡]*, Steven M. Roche[1,3,‡], Stephen J. LeBlanc[1,‡]

1 Department of Population Medicine, University of Guelph, Guelph, Ontario, Canada, 2 Department of Public and Ecosystem Health, Cornell University College of Veterinary Medicine, Ithaca, New York, United States of America, 3 Agricultural Communications & Epidemiological Research (ACER) Consulting Limited, Guelph, Ontario, Canada

◉ These authors contributed equally to this work.
‡ CCA, SMR and SJL also contributed equally to this work.
* ccobo@uoguelph.ca

**Data Availability Statement:** All relevant data from the quantitative study can be found within the paper and its Supporting information files. Raw

## Abstract

The aim of this study was to describe the factors that influence dairy cattle veterinarians´ antimicrobial prescribing, their attitudes toward reducing antimicrobial use (AMU) in the dairy industry, awareness of antimicrobial resistance (AMR), and perceived barriers to improving antimicrobial stewardship (AMS) on Canadian dairy farms. We used quantitative and qualitative research methods to consider the complexity of the antimicrobial prescription decision-making process. We designed and distributed an online survey, followed by four online focus groups with practicing veterinarians. We used frequency tables and unconditional associations to analyze quantitative data, and thematic analysis to analyze qualitative data. In total, 107 participants from four Canadian provinces responded to the survey, and 26 veterinarians participated in the focus groups. Results from both studies were triangulated to draw our key findings. We found that 1) Veterinarians must weigh numerous intrapersonal and contextual considerations that could be conflicting in their decision-making process for prescribing antimicrobials. 2) Although less experienced veterinarians showed greater awareness of AMR and motivation to improve AMS than more experienced veterinarians, they also reported feeling more pressure to adapt their prescribing practices to farmers' wishes than more experienced veterinarians. 3) Some veterinarians experienced conflict between prescribing antimicrobials to maintain animal health, productivity, and welfare, and AMS, which could result in blaming others for inappropriate antimicrobial use and reducing the opportunity to critically review their own prescribing practices. 4) There were strategies and barriers in common between veterinarians and farmers such as collaborative reviews of disease treatment protocols and improving preventive medicine on dairy farms. 5) The main barriers to reducing AMU on dairy farms reported by veterinarians were concerns about animal welfare and AMU on dairy farms without consultation with the veterinarian. Our results can inform the development of AMS programs in the Canadian dairy industry.

data is publicly available at https://doi.org/10.5683/SP3/VBFGOX. However data from qualitative study cannot be shared publicly because it contains sensitive information from qualitative interviews. Full interview transcripts cannot be shared due to the terms of the participants' consent. Some transcript excerpts are included in the text of the paper Data availability for research purposes will be evaluated on a case-by-case basis through the Research Ethics Boards (reb@uoguelph.ca).

**Funding:** This research was funded by the University of Guelph Food From Thought research program (Grant 499104). The funders had no role in study design, data collection and analysis, decision to publish, or preparation of the manuscript.

**Competing interests:** The authors have declared that no competing interests exist.

# 1. Introduction

Antimicrobial-resistant infections are a substantial public health challenge [1]. Disease caused by resistant bacteria results in increased severity, prolonged hospital stays, and increased mortality risk, raising the social and economic costs of the disease [2]. The development of antimicrobial resistance (AMR) is part of normal bacterial evolution [3]. However, antimicrobial use (AMU) in humans and animals contributes to the development and spread of AMR [4, 5]. Therefore, health authorities have called for antimicrobial stewardship (AMS) in human and veterinary medicine to ensure prudent AMU and minimize the selection and dissemination of resistant bacteria [6, 7].

Several actors play a role in AMS in food-producing animals, including regulators, farmers, and researchers. In addition, veterinarians are commonly central to antimicrobial stewardship on farms. For example, in the Netherlands, only veterinarians are authorized to administer antimicrobials on farms, with exceptions when a one-on-one relationship between a farmer and a veterinarian is established and documented [8]. In the United States, veterinarians have the obligation of oversight for use of medically important antibiotics in food-producing animals and to ensure that these antimicrobials are only used when medically necessary to protect the health of an animal [9]. In Canada, all Medically Important Antimicrobials (MIAs) for veterinary use are sold by prescription only [7]. Therefore, it is necessary to understand the factors that influence veterinarians' antimicrobial prescribing decisions and their attitudes toward reduced AMU.

Numerous organizations have developed AMS guidelines for veterinary medicine. In 2016 the World Health Organization (WHO) in collaboration with the World Organization for Animal Health (WOAH) and the Food and Agriculture Organization of the United Nations (FAO), adopted a Global Action Plan with the objective of ensuring treatment and prevention of infectious diseases with quality-assured, safe, and effective medicines [6]. The WOAH included guidelines for responsible AMU in veterinary medicine as part of the Terrestrial Animal Health Code [10]. The Public Health Agency of Canada released a Federal Action Plan on AMR and AMU, which established promotion of the appropriate use of antimicrobials in human and veterinary medicine as part of the actions needed to tackle AMR. Consequently, the Canadian Veterinary Medical Association (CVMA) published Veterinary Oversight of Antimicrobial Use with recommendations and standards for responsible antimicrobial prescribing and dispensing [11]. The CMVA recently launched an online platform intended to support Canada's veterinarians in making responsible decisions on appropriate AMU (https://www.canadianveterinarians.net/veterinary-resources/antimicrobial-stewardship-resources/cvma-guidelines-for-veterinary-antimicrobial-use/). Nonetheless, the development of AMS recommendations and guidelines does not ensure their implementation. For instance, a study from Australia reported that only 15% of veterinarians used antimicrobial prescribing or AMS policies [12]. Companion animal veterinarians described conflicts in prescribing antimicrobials using current guidelines due to a lack of time to keep track of current recommendations and lack of autonomy at the clinic [13].

Prescribing decisions are multifactorial, with geographic, social, cultural, and economic factors playing significant roles [14, 15]. It is necessary to study the interaction between individual and contextual factors to understand their influences on prescribing practices [14]. Contextual factors influence how veterinarians will act in specific situations; these include legislation, working experience, farmers' and colleagues' preferences, and the structure of veterinary practices, among others [16]. Personal experiences and emotions guided dairy cattle veterinarians when making AMU choices and less experienced veterinarians felt pressure to prescribe according to farmers' preferences for certain antimicrobials [16]. A study from the

Netherlands identified conflicting interests when veterinarians prescribed antimicrobials to farm animals, such as the belief that veterinarians have the obligation to alleviate animal suffering, financial dependence on clients, financial barriers for structural veterinary herd health advisory services, and farmers' lack of compliance with veterinary recommendations [17].

Our objective was to describe the perceptions and influences in AMU decision-making by dairy veterinarians. Considering the complexity and multifactorial nature of veterinarians' prescribing decisions, we conducted mixed methods research to specifically explore dairy cattle veterinarians' considerations for antimicrobial prescribing, their attitudes toward reducing AMU, awareness of AMR, and perceived barriers to improving AMS on dairy farms in Canada.

## 2. Methods

We used multistage mixed methods with an online survey questionnaire followed by online focus groups. This research was reviewed and approved by the University of Guelph Research Ethics Board (documents #21-02-021 and #21-10-025). Online informed consent was obtained from participants before both parts. All participants were informed about the study objectives, methods, and implications for them and for the field of study. Respondents and focus group participants were required to be at least 21 years old to participate. All participants were aware of their right to choose not to answer any questions. Survey participants consented to participate in the first page of the survey and focus group participants' consent was collected online two days before the activity together with demographic information.

### 2.1 Quantitative study

**Survey design and distribution.** We designed a questionnaire based on the literature and our experience. The questionnaire was pre-tested by four dairy cattle veterinarians for comprehension and refined based on their feedback. The survey contained 42 questions related to general information on the respondents (4 items), antimicrobial prescribing decisions (8 items), AMU and AMR awareness (19 items), and AMU reduction (11 items). Several questions about AMU and AMR were the same as those in our recent survey of dairy farmers in Ontario [16]. The questionnaire is available in Supplementary Materials (S1 Appendix). The survey was distributed using the Qualtrics XM software (Qualtrics, Provo, UT, USA). Participants were recruited by email through email lists of the Canadian Association of Bovine Veterinarians (690 subscribers). Reminders were sent 2 weeks and one month after initial recruitment. The survey link was open from February 7, 2022 to April 20, 2022.

**Data analysis—Quantitative study.** We used descriptive statistics and frequency tables to explore the distribution of the data. Unconditional associations between variables were tested using Pearson Chi$^2$ or Fisher's exact tests for categorical variables. For continuous data, t-tests or Wilcoxon rank–sum tests were used to explore differences between categories when the continuous variable was or was not normally distributed, respectively. All analyses were conducted using the software Stata 17 (StataCorp, College Station, TX, USA).

### 2.2 Qualitative study

We developed a semi-structured interview guide based on the results from the quantitative study and the literature, including 19 open-ended questions. Questions and probes sought information on drivers for antimicrobial prescribing, external influences on antimicrobial prescribing, AMR awareness, and attitudes toward AMU reduction on dairy farms. The interview guide is included in the Supplementary Material (S2 Appendix).

**Table 1. Demographics of participants in quantitative (online survey; n = 88) and qualitative (online focus groups; n = 26) studies of antimicrobial prescribing decisions by Canadian veterinarians working with dairy farms.**

| Characteristic | Quantitative study | Qualitative study |
|---|---|---|
| Pronoun used | | |
| He/Him | 59 (63%) | 14 (54%) |
| She/Her | 35 (38%) | 12 (46%) |
| Age median (range) | 44 years (28 to 69) | 36 years (29 to 69) |
| Median years of experience working as a dairy cattle veterinarian (range) | 17 (2 to 48) | 10 (1 to 43) |
| Current position | | |
| Owner or partner in a private veterinary clinic | 45 (48%) | 15 (58%) |
| Associate in a private veterinary clinic | 36 (39%) | 10 (38%) |
| Working for an academic institution | 6 (6%) | 1 (3%) |
| Other work positions | 6 (6%) | |
| Province of residence | | |
| Ontario | 57 (64%) | 26 (100%) |
| Quebec | 13 (14%) | |
| Alberta | 9 (10%) | |
| Saskatchewan | 4 (5%) | |

We conducted four online focus groups using the Zoom platform (Zoom Video Communications, Ca, USA) between April 12, 2022, and April 26, 2022. All the participants resided and worked in the province of Ontario, all primarily as veterinarians in dairy cattle practice. All focus groups were moderated by one of the authors (SMR) accompanied by the first author, who was present taking field notes. Survey respondents who resided in Ontario and agreed to be contacted for further studies were invited to participate in the focus groups. Additionally, veterinarians were invited to participate by email through the Ontario Association of Bovine Practitioners. In total, 26 veterinarians participated in the focus groups. Each focus group was held with four to eight participants and lasted 1.5 hours. Before the focus group, participants completed a short online questionnaire about their general information (Table 1). All focus groups were audio and video recorded and transcribed by a professional transcriptionist.

**Data analysis—Qualitative study.** Thematic analysis was used to identify, analyze, and report patterns within the written data using NVivo software (QSR International Pty Ltd., 2022). The first author (CC-A) read the transcriptions and conducted an initial coding using a bottom-up, inductive approach aimed at identifying potential themes and subthemes from codes assigned to text elements that informed the research objectives. Subsequently, the authors discussed the codebook, themes, and subthemes. Adjustments were made until a consensus was reached. We assigned an anonymous ID to each participant and assigned codes to the quotes to illustrate the key features of themes and subthemes. Square brackets were used to indicate when a quote was shortened or when we inserted explanatory information to ensure the meaning of the quote was maintained.

Findings from the qualitative and quantitative study were triangulated and are presented in an integrated form in the results session.

## 3. Positionality and reflexivity statement

We acknowledge that the researcher's positionality impacts the way that data are generated and analyzed [18]. All researchers of this study have professional knowledge of dairy farming, two are veterinarians, and all have advanced degrees in epidemiology. The last author (SL) was

acquainted with almost all the participants in the focus groups. However, he did not participate in those sessions. The first author (CC-A) led the data analysis and manuscript writing. CC-A is a female postdoctoral researcher, whose work focused mainly on AMR and AMS in dairy farms with a One Health approach. Differences in sociocultural background between CC-A and the participants arise due to the fact that CC-A is not Canadian and has never practiced veterinary medicine in Canada. However, as a veterinarian, CC-A understands the role of AMU in dairy cattle health and production and the complexity of the prescribing decisions. She has spent the last three years conducting qualitative and quantitative research with Canadian dairy farmers and veterinarians aimed to improve AMS on dairy farms.

## 4. Results

### 4.1 Demographics

Of 107 veterinarians who used the survey link, 93 responded to more than 50% of the questions and 88 respondents completed the full questionnaire. Most respondents identified themselves with the pronoun He/Him (n = 59; 63%), resided in the province of Ontario (n = 57; 65%) were owners or partners in a private veterinary clinic (n = 45; 48%), and reported having prescribed antimicrobials in more than half of their last 10 visits to dairy farms (65%; n = 59). A description of the participants in both parts of this study is presented in Table 1. About half of the participants in the focus groups identified themselves with the pronoun He/Him (n = 12; 54%), and most of the participants were owners or partners in a private veterinary clinic (n = 15; 57%).

### 4.2 Key themes identified

We identified three key themes in our data that are described in Fig 1. Broadly speaking, the data highlight that antimicrobial prescribing decisions are driven by intrapersonal factors such as education and experience, as well as external factors related to the farm, drug availability, animal value, and current regulations, among others. Veterinarians must weigh numerous considerations that could be conflicting in their decision-making process for prescribing antimicrobials to their clients. A comment from one focus group participant captures the key aspects of the themes identified [P7_4]: *"Every single case that needs antibiotics, I try to pick as low class [category of importance for human medicine] as possible. Try to pick something as effective as possible. And try to pick something that the farmer is going to be able to implement"*.

**Theme 1: Decision-making based on age education, experience, and role in the clinic.** The first considerations that veterinarians expressed in our focus groups were related to personal factors that influence their prescribing decisions. The following sub-themes were identified:

*a. Veterinary knowledge of infectious diseases and drug efficacy*:

The most important factor in selecting a particular antimicrobial product by survey respondents was "Knowledge of antimicrobial drug efficacy to treat specific diseases". Similarly, knowledge and experience with the antimicrobial was the first consideration discussed by focus group participants when they were asked about how to decide whether or not to prescribe for a given case (Table 2; S2 Fig in S1 File).

Additionally, veterinarians considered diagnosis and specific signs and symptoms as triggers of antimicrobial prescribing, including fever and decreased milk or feed consumption (Table 2; S2 Fig in S1 File). Although most focus group participants admitted that those signs and symptoms are not conclusive for the need for antimicrobials, participants frequently mentioned that they preferred to be cautious and treat an animal before the animal deteriorates or

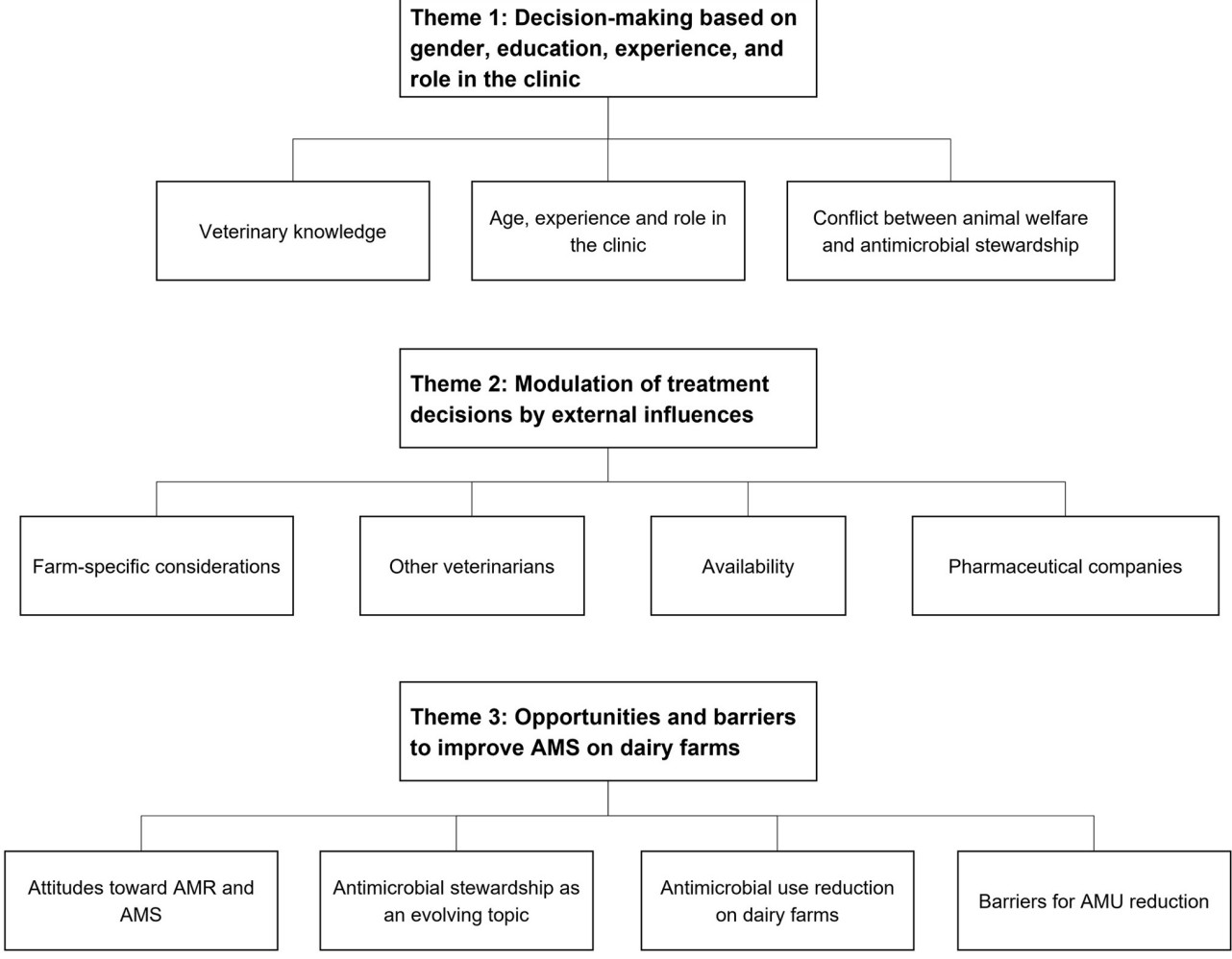

**Fig 1. Thematic map of the analysis of four focus groups of veterinarians on antimicrobial prescribing decisions and attitudes toward antimicrobial resistance and antimicrobial stewardship on dairy farms in Canada.**

even dies while waiting for a precise diagnosis. Veterinarians commented on the situations in which they recommend extra-label antimicrobial use (i.e., use of an approved drug in a manner that is not in accordance with the approved label directions). The most common extra-label use discussed was the extended duration of antimicrobial therapy. Participants considered that severe cases of mastitis and pneumonia usually benefit from extended treatment, as well as cases that didn't respond to the initial treatment protocol.

Some participants commented on the use of diagnostic tests such as lung ultrasounds, nasal swabs, and milk culture. However, it was commonly mentioned that microbiological cultures were used as historical information on herd health status rather than as a guideline for treatment. Many veterinarians also reported that susceptibility tests were almost exclusively used for recurring or severe mastitis cases. Some participants questioned the value of performing a susceptibility test when there are limited antimicrobial options for intramammary treatment (Table 2).

*b. Gender, age, experience, and role in the clinic*

**Table 2. Description of qualitative and quantitative results related to theme 1: Decision-making based on gender, education, experience, and role in the clinic.**

| Theme and subthemes | Quote(s) reflecting the theme or subtheme | Quantitative results supporting the theme or subtheme |
|---|---|---|
| **Theme 1: Decision-making based on gender, education, experience, and role in the clinic** | | |
| a. Veterinary knowledge of infectious diseases and drug efficacy | [P8_2]: *"...it has to be a case that warrants it [antimicrobial treatment], based on the health issues that the animal has. So, the antibiotic is necessary, and then we would look for an antibiotic that would be effective".*<br><br>[P10_2] *"I err on the side of treating even though I'm sure the odd one isn't truly pneumonia. Part of my reasoning is that you were called out to see them. They're obviously quite ill. I'd rather err on the side of caution in that sense for the cow. Especially when the producer deems that severe enough for a vet call".*<br><br>*Moderator*: in what situations do you perform a bacteriologic culture and susceptibility test? [P2_1]: *"the obvious response is mastitis. But it is of limited use when you have so few treatment options. It is a hard sell for some producers when you know they know there are only two lactating cow intramammary treatments on the shelf. So, a lot of cases do not get cultured".* | 86% (n = 74) of respondents considered that knowledge about the antimicrobial was an extremely or very important factor to decide the specific product to prescribe (S1 Fig in S1 File).<br><br>98% (n = 82) considered diagnosis as extremely or very important factors related to the animal considered in the decision-making process to select an antimicrobial treatment (S1 Fig in S1 File).<br><br>60% (n = 49) of respondents affirmed not having performed a microbiological culture in more than half of the cases they attended in the last year.<br><br>35% (n = 28) of respondents did not perform a microbiological culture in the last year.<br><br>24% of the veterinarians who performed a microbiological culture never requested a susceptibility test and 48% (n = 37) performed susceptibility testing in less than half of microbiological cultures in the last year. |
| b. Gender, age, experience, and role in the clinic | [P22_4]: *"...as a new grad, I would certainly struggle to convince them [farmers] you know the science doesn't suggest we should use antibiotics on a scouring calf. I think that they're probably expecting me to do what the old Doc did. I guess I'm more inclined to use something [antimicrobial] in those cases".*<br><br>[P16_3]: *"Rebates matter as the antibiotic choices between companies, right? Kind of choose your brand and you might stock the one that has the better price. But in terms of I'm going to stock A-180 because it's a better price than penicillin? Like that wouldn't enter my mind".* | 46% of veterinarians with less than 10 years of experience reported performing a microbiological culture in less than 5% of the cases that they attended, which was less than more experienced practitioners (46% vs 18%; P = 0.02).<br><br>Considering peer-reviewed papers on efficacy as a "very" or "extremely important" factor when prescribing was different between pronouns (34% among She/her and 21% among He/him; P = 0.04).<br><br>Respondents with the pronoun He/Him considered farmers' goals more important than respondents with the pronoun She/Her (68% vs. 32%; P = 0.005). |
| c. Conflict between animal welfare and antimicrobial stewardship | [P9_2] *"...I think in many cases we are the voices for the animals under our care. And often the only ones who advocate for them, you know? And many times, there are certain circumstances where the issues that cause the outbreak are challenging. And the most humane thing to do and animal welfare is better addressed by doing across the board treatment".*<br><br>[P10_1] *"I had a situation with the 4-6-month-old heifers, and they [farmers] pulled two dead ones out of the pen. Classic BRD [Bovine Respiratory Disease] outbreak and I treated the whole group— there were 30 of them. Because to me, yes, we need to be prudent with our antibiotic usage. But to me, it's like we also have to treat the animal. And we have the ability to save them and keep them functioning".*<br><br>P1_1: *"We made a conscious choice not to sell category I dry cow therapy. Although we may eventually be forced to because there's almost no mastitis preparations left at all on the market".* | 43% (n = 37) of respondents considered that animal welfare would be worse if AMU is decreased. |

Veterinarians discussed that their age and role in the clinic could affect their prescribing decisions. For instance, younger participants expressed that it is difficult to suggest changes to farmers that conflict with recommendations from more experienced veterinarians. Additionally, some veterinarians expressed that being a clinic owner adds pressure to the economic considerations of antimicrobial prescribing. Clinic owners agreed that their role does not influence the decision of whether to prescribe or not, but it could influence the decision of

which brand of the same antimicrobial to prescribe based on the clinic's relationships with pharmaceutical companies.

*c. Conflict between animal welfare and antimicrobial stewardship*

In this study, veterinarians expressed that they consider themselves stewards of animal welfare, and that responsibility was highlighted when participants discussed prophylactic antimicrobial use (Table 2). In addition, many participants expressed that they try to avoid the use of category one (high importance to human medicine) antimicrobials due to concerns about AMR in the human population (Table 2; S2 Fig in S1 File). However, it was commonly mentioned that the responsibility for maintaining animal health and the responsibility for AMS conflicted. Other participants expressed that although they would like to avoid category one antimicrobials, they are forced to use them because of the lack of availability of other products (Table 2).

**Theme 2: Modulation of treatment decisions by external influences.**   This theme relates to the external factors that are also considered when veterinarians prescribe antimicrobials to their clients. External factors discussed in the focus groups included:

a. *Farm-specific considerations*

Veterinarians commented that some farm characteristics and management were considered at the moment of prescribing antimicrobials. The milk production status regarding the milk quota (i.e., daily kg of butterfat that each farmer is allowed to produce) of the farm was an important factor for many veterinarians when selecting the type of product (Table 3). Another factor was the housing conditions. Some participants commented that if the calf or heifer housing did now allow proper ventilation, they would be more likely to recommend medicated feed. In addition, veterinarians agreed that they usually take into consideration the farmers' preferences and requests when they are selecting a specific product in terms of the cost of the treatment. Most participants commented that they try to follow the existing protocols on the farms, and they consider the farmer's expected compliance with the recommendations (Table 3).

b. *Other veterinarians*

Most veterinarians commented that they have regular meetings with other veterinarians from their clinic to discuss protocols, treatment choices, and difficult or unusual disease cases in order to offer similar procedures to their clients (Table 3). Other participants commented that they are interested in reviewing protocols developed by other veterinarians to learn about how they treat diseases. Less experienced participants expressed that they feel more influenced by older veterinarians and mentors and admitted that most of their prescribing practices are learned from other veterinarians. In addition, some participants considered that younger veterinarians are more informed about AMS and are more concerned about the prudent use of antimicrobials.

c. *Availability*

It was frequently mentioned that antimicrobials have been disappearing from the market in the last few years, limiting the options for veterinarians (Table 3). In addition, participants expressed that their veterinary clinic sometimes limits which antibiotics they carry in order to improve antimicrobial stewardship (Table 3).

d. *Pharmaceutical companies*

Veterinarians' opinions on how pharmaceutical companies influence their prescribing practices were divided. A group of participants considered that sales or technical services

**Table 3. Description of qualitative and quantitative results related to theme 2: Modulation of treatment decisions by external influences.**

| Theme and subthemes | Quote(s) reflecting the theme or subtheme | Quantitative results supporting the theme or subtheme |
|---|---|---|
| **Theme 2: Modulation of treatment decisions by external influences** | | |
| a. Farm-specific considerations | [P26_4] *"I have to work with the farmer just to see if he needs the milk or something then we can maybe go to one of those ones that have low or zero milk withdrawal".*<br><br>[P9_2] *"...At the end of the day, I will recommend what I think they [farmers] will do, what will get done. Because you know there are times when that is probably of greatest importance you know? There's a variety of options, but this one will get done you know then that goes to the top of the list for that client".* | 82% (n = 69) of respondents considered the expected compliance with the treatment as an extremely or very important factor, related to the farm when deciding which product to prescribe (S1 Fig in S1 File). |
| a. Other veterinarians | [P6_1] *"We'll occasionally have a meeting where we'll bring up why we do things the way we do things and see if we can change each other's minds essentially. So people can be on the same page [...] We do enjoy being all on the same page. So that the client is getting a similar experience".* | 52% (n = 35) of veterinarians considered that colleagues' reports of efficacy is an extremely or very important factor to select a particular antibiotic product. |
| b. Availability | [P16_3] *"It's frustrating, it seems like the antimicrobials are going away are in the lower drug class, so we're getting smaller and smaller options, but it's not the category ones that are going away. It's the category threes and fours. And it's forcing us to have to use those category ones because two options for mastitis treatments that are on the market".*<br><br>[P7_2]: *"We don't carry fluoroquinolones anymore; if the farmer asks for it, I'm sorry I just don't have any with me right now".* | |
| c. Pharmaceutical companies | [P26_4]: *"I think they [sales representatives] are useful getting those tech service bulletins, getting some advice, getting some newsletters things like that, information. Always being available to us when we need them. If there's some sort of problem that's arisen from one of their products, maybe not always compensation. But at least support".*<br><br>[P21_4]: *"...these [pharmaceutical] companies are so based on marketing promotion and lack of research, and the data behind their product is based on testimonials. I just don't deal with them".* | 81% (n = 67) of respondents considered marketing material from pharmaceutical companies not at all important in the decision of which antimicrobial product to prescribe. |

representatives provide updated information about treatment length, doses, and availability of antimicrobials (Table 3). However, other participants considered that information offered by pharmaceutical companies is not reliable and do not consider them influential (Table 3).

**Theme 3: Opportunities and barriers to improving AMS on dairy farms.**

*a. Attitudes toward AMR and AMS*

Participants had divided attitudes toward public discussions of AMR and AMS in dairy farming. While some veterinarians expressed being concerned about AMR and feeling responsible for reducing AMU on dairy farms, other veterinarians commented that AMU in human medicine is responsible for AMR and questioned the link between AMU in dairy cattle and the emergence of AMR in human pathogens. In addition, participants considered that AMU in dairy cattle is being more scrutinized than AMU in other animal species or human medicine. Some participants expressed frustration because they consider that veterinarians try to use antimicrobials responsibly, but human medicine is not doing its part (Table 4). Finally, another group of participants considered that they do not have enough information to discuss the role of AMU on dairy cattle on AMR. The level of agreement of veterinarians with 18 statements related to antimicrobial resistance awareness in the quantitative part of the study is presented in S2 Fig in S1 File.

**Table 4. Description of qualitative and quantitative results related to theme 3: Opportunities and barriers to improving AMS on dairy farms.**

| Theme and subthemes | Quote(s) reflecting the theme or subtheme | Quantitative results supporting the theme or subtheme |
|---|---|---|
| **Theme 3: Opportunities and barriers to improving AMS on dairy farms** | | |
| a. Attitudes toward AMR and AMS | [P4_1]: *"I think that in dairy cattle, for the most part, we do use drugs appropriately, and with you know the right length of treatment, and the right you know, dosages and administration means. And It's been a couple of years now since Quebec has changed their rules. And I would love to see if they've seen any changes in their patterns of resistance. But I don't think dairy contributes a ton".*<br><br>[P14_3] *"when I look in the small animal pharmacy there's some pretty good category I drugs there that are routinely stocked, and I don't hear a lot of conversation about that. And it frustrates me when I go to the doctor and they prescribe me something and they can't calculate a mg per kg dose. We can do that. Math isn't that hard".*<br><br>[P11_3] *"I think we [veterinarians] get blamed. Agriculture gets blamed, and maybe the piece of the pie is bigger. I don't think we're a big problem. And then I, you know go to my doctor, my kids go to the doctor, and they don't even examine me or do anything. And they're handing antibiotics. And it's just frustrating, and I think sometimes we're doing a better job than our counterparts in human medicine".*<br><br>[P5_1]: *"I don't usually talk about it [AMR] with people because I feel like I really don't know enough. How can I have an opinion on something that I feel like I really don't know enough?".* | 62% (n = 51) of respondents consider that veterinarians generally prescribe responsibly.<br><br>50% (n = 41) of respondents disagree or strongly disagree that veterinarians are responsible for AMR in animals.<br><br>60% (n = 82) disagree or strongly disagree that veterinarians are responsible for AMR in human medicine.<br><br>64% (n = 52) of veterinarians considered that reduced AMU will not affect milk production.<br><br>54% (n = 44) of respondents were unsure of the impact of reduced AMU on animal welfare.<br><br>61% (n = 50) of respondents think about the risk of AMR in cattle when they prescribe.<br><br>50% (n = 51) of respondents think about the risk of AMR in humans when they prescribe.<br><br>95% of veterinarians consider AMR as an important problem in human medicine. |
| b. Antimicrobial stewardship as an evolving topic | [P3_1]: *"I think we're a lot more prudent with antimicrobial usage now than we ever have been in the past. And I think a lot of that is due to what we hear, and what their [public] expectations are. And antimicrobial resistance is a big part of that. I know I use a lot fewer drugs now, and extended drugs, and I'm not over-treating as much as before".*<br><br>[P10_2]: *"They [farmers] are changing, it might not be quick, but it does happen. Especially as new generations take over. I've really seen that the newer generation really wants your advice, wants to make changes, wants to hear more about science. I think that's how we use less antimicrobials, and we have more prudent use it is a slow change, but is still change".* | 93% (n = 76) of respondents considered that there should be more initiatives to promote responsible AMU in the dairy industry.<br><br>72% (n = 60) of respondents agreed or strongly agreed that we should reduce antimicrobial use in dairy production.<br><br>90% (n = 73) of veterinarians agreed or strongly agreed that they avoid prescribing critically important antimicrobials for human medicine. |
| c. Antimicrobial use reduction on dairy farms | *"P1_1: The area on the farm where I feel like we could do a better job is calves. I think that there's a lot of calves out there who get antibiotics randomly because they're small and it's not as expensive".*<br><br>[P19_4]: *"A lot of them [farmers] aren't at this point going to make a decision about drug use based on responsible use. It's definitely going to be more economically driven. So if there was some way to back up these recommendations, and say look this is actually more profitable and better for your animals that would be really helpful".*<br><br>[P27_4] *"I think one of the biggest opportunities for responsible use is getting the right diagnosis, sometimes our clients are treating animals, where it's not indicated. So I guess, that opportunity is just, you know, educating our clients better to make sure that they're treating when they should be treating with the right antibiotic".* | 40% (n = 33) of respondents regarded dry cows as the group of animals in which it is easier to reduce AMU, followed by pre-weaned calves (19%; n = 16) and lactating cows (16%; n = 12).<br><br>54% (n = 48%) considered that veterinarians should take the lead on promoting responsible AMU in the dairy industry, and 255 (n = 22) considered that dairy farmers should be responsible for promoting responsible AMU on dairy farms.<br><br>Among the actions considered by veterinarians to be most effective to reduce AMU on dairy farms, "Measure use and provide benchmarking to compare use among dairy farms" was the most frequently selected (36%; n = 30), followed by "Provide more education or promotion to change or reduce use" (26%; n = 22), and "Provide incentives to change or reduce use" (21%; n = 18).<br><br>Less than 10% (n = 6) of respondents considered that additional regulations would be effective to reduce AMU on dairy farms. |

(*Continued*)

**Table 4.** (Continued)

| Theme and subthemes | Quote(s) reflecting the theme or subtheme | Quantitative results supporting the theme or subtheme |
|---|---|---|
| d. Barriers for AMU reduction | [P16_3]: *It's frustrating how many times you get called and they've had them [animals] on Excenel [ceftiofur] for two days, and there was like did you check for a fever? No? Well, why did you use Excenel? Well, it worked on the last cow".*<br><br>[18_3]: *"We haven't figured out how to manage our geography in our area to make an actual mastitis culture lab feasible. And so, I'd say that's a huge barrier for us to managing some of these cases better. And having better antibiotic use. This space between our farms makes culturing mastitis cases well completely impractical for the most part because it's at least a week turnaround. Like no one's going to drive to bring us a milk sample to a clinic that is 40 minutes from the house with nobody there staffing on the weekend to manage a culture lab".*<br><br>[P9_2]: *"I know from experience when vets are not on the same page within a practice that there's a lot of client confusion and they want you to do it the way someone else did it even though that's maybe not the way you want to practice. But then I do think other clinics put pressure on their prescribing on other local clinics, right?".*<br><br>[P18_3] *"Just because I work across multiple species one of the things I notice is that a lot of information for small animal tends to be very directed at messaging, they tend to take research and turn it into bite-sized information that we can educate clients. And I find that in the bovine world they tend to give us research without actually being able to synthesize and coordinate it into a very simple message for farmers".*<br><br>[P25_4]: *"It [AMR] is intangible to a lot of dairy producers. They don't really see the direct impacts of what antimicrobial resistance could have on them and the class of antimicrobials is the last thing on their radar".* | 51% (n = 42) of veterinarians in this study considered that their clients use antimicrobial treatments without consultation in at least 80% of the disease cases on their farms. However, the majority (81%; n = 66) of respondents affirmed to have had a discussion with their clients about when and how to use antimicrobials for treatments when they are not being consulted.<br><br>53% (n = 43) agreed or strongly agreed that farmers have limited understanding of AMR. |

*b. Antimicrobial stewardship as an evolving topic*

Most participants agreed that AMS is a relatively new concept for them, that the concerns of AMR and prudent prescribing are evolving, and they have had to adapt their practices and protocols to comply with new requirements (Table 4). Additionally, veterinarians considered that the current discussion on AMR and AMU in agriculture has increased farmers' awareness, particularly younger farmers, who are now more willing to modify their traditional practices of AMU (Table 4). In the quantitative part of the study, veterinarians with less than 10 years of experience were more likely than more experienced practitioners to agree or strongly agree with the following statements: "Antimicrobial resistant infections are an important problem in dairy cattle" (P = 0.05), "The use of antimicrobials on a farm could cause antimicrobial resistance on that farm" (P = 0.04), "The use of antimicrobials on dairy farms could cause antimicrobial resistance on other farms" (P < 0.01), and "Antimicrobial resistance is an issue that could affect me or my family" (P = 0.05).

*c. Antimicrobial use reduction on dairy farms*

Although there was a consensus that dairy farming generally uses antimicrobials prudently, veterinarians expressed several ideas to reduce AMU on dairy farms. The most common ideas were to implement selective dry cow therapy, improve the diagnosis of mastitis, and have better culling protocols for animals with recurrent infections. Participants agreed that calves were a particular age-class of animals where there is opportunity to further reduce AMU (Table 4).

Improving vaccination protocols was also an idea expressed by several participants as potential routes for reducing AMU in dairy farms. Furthermore, it was proposed that a technician in the clinic could support farmers with vaccination to ensure that they are being used properly.

It was also discussed that prudent use of antimicrobials may not be enough motivation to achieve reduced AMU on farms and that the discussion about reducing AMU should be driven by economics. Other veterinarians proposed having grants to improve the farm facilities to have better ventilation, calf housing, and animal welfare. Participants also discussed their role in AMS and mentioned that they could be more involved in AMU decisions; for example, reviewing protocols for disease treatments to identify areas for improvement and acknowledged that disease diagnosis could improve in order to reduce AMU (Table 4)

*d. Barriers for AMU reduction*

Veterinarians expressed frustration about farmers' habit of treating the animals without consultation, and this practice was considered one of the biggest barriers to reducing AMU on dairy farms in Canada (Table 4). All the focus groups discussed the impediments to use of laboratory services for disease diagnosis due to the long time and distance between farms and laboratories, which causes delays for results. Another barrier mentioned by participants was differences between veterinarians and between clinics that put pressure to prescribe what farmers demand (Table 4). In addition, participants discussed that knowledge mobilization to dairy cattle veterinarians and farmers is given in technical language that is difficult to understand. Finally, veterinarians considered that farmers had low awareness of AMR implications in dairy farming, which they considered a barrier to reducing AMU on dairy farms. Other participants expressed that the poor adoption of preventative measures to reduce diseases on some farms is an important barrier to improving AMS.

## 5. Discussion

We used a quantitative survey and qualitative focus groups to study dairy cattle veterinarians' antimicrobial prescribing decisions, AMR awareness, and attitudes toward AMU reduction in the dairy industry. Mixed methods research has the advantage of allowing examination of complex problems within a single study by providing multiple sources of evidence [19]. Our results show that antimicrobial prescribing decisions are complex for veterinarians, who need to consider a number of personal and external factors that are connected; in most cases, it is difficult to classify a specific factor as exclusively personal or external. For instance, age, gender and experience are usually considered individual factors. Nevertheless, they are related to how veterinarians process external influences and the way that context affects their decisions. We identified that participants with the pronoun She/Her were more likely to report avoiding the prescription of critically important antimicrobials, while participants with the pronoun He/Him were more likely to consider the farmer's goals in the prescribing decision. Social sciences have described that there are gender differences in prioritization of values [20], in which women tend to put more value on protecting the welfare of all people, animals, and nature, while men attributed more value to social status, demonstrating competence and obtaining admiration [20]. This illustrates how personal traits affect information processing and decision making, and needs to be considered to understand the required support, advice, and guidance to modify any type of behavior, including prescribing decisions.

Similarly, we found that younger veterinarians felt more influenced by farmers and other veterinarians, affecting their prescribing decisions. Although less experienced participants were more likely to agree or strongly agree with statements reflecting awareness of AMR, they also reported performing microbiological cultures and susceptibility testing less frequently.

This contradictory finding may be related to the farmers' preference for rapid AMU for fear of an adverse clinical outcome or economic losses while the animal is sick, as reported in previous studies [21–24]. This situation reflects the complex dynamic between individual and contextual factors. Despite greater awareness of AMR and motivation to improve AMS, younger veterinarians chose to adapt their prescribing practices to farmers' and more experienced veterinarians' preferences or practices. Previous research has also shown the complexity of social influences on antimicrobial prescribing behavior. Sometimes veterinarians prescribe antimicrobials when they feel that it is not necessary due to pressure from their clients [25]. This has also been reported in human medicine [26]. The importance of social influences underlines the need to design AMS interventions that involve all stakeholders, including farmers and veterinarians. Changing the external factors, in this case, the social pressure of antimicrobial prescribing, could affect individual behavioral change and improve AMS [25].

Increasing awareness and understanding of AMR is one of the key strategies of the Global Action Plan on AMR adopted by the WHO, FAO, and WOAH [6]. In the present study, more than 90% of survey participants considered AMR to be an important problem in people. However, less than half of the participants (42%) considered that AMR is a problem in dairy cattle. We did not specify in our question what type of problem AMR represents for dairy cattle. We intended and expect that veterinarians took this question to mean a problem for animal health such as not being able to treat resistant infections, but they may have considered it a problem for AMR transmission to other animals, humans and the environment. In any case, most veterinarians did not consider AMR a problem, even when focus group participants frequently discussed having to deal with persistent or recurring infections. This might imply low recognition of AMR in daily practice. Although clinical failure does not necessarily mean an AMR infection, some dairy cattle pathogens feature a high prevalence of AMR, including mastitis pathogens such as *Staphylococcus aureus*, and *Klebsiella* [27] and diarrhea-causing bacteria such as *E. coli* and *Salmonella* [28]. Conversely, participants may accurately see few dire clinical outcomes or treatment failures attributable to AMR in dairy cattle. This dissonance was also seen in the high proportion of participants who agreed that they try to limit use of antimicrobials that are critically important for human health, but responses were more neutral about considering the effect of their AMU on AMR in humans. The former response might reflect social desirability while the latter may reflect ambivalence about a tangible link between their actions and AMR and people. Similarly, focus group discussions revealed that some veterinarians considered that they are not responsible for AMR and expressed that the AMR discussion should be focused on AMU in human medicine. Although the link between AMU on dairy farms and AMR in humans has not been quantified, there is growing evidence supporting the association between AMU on dairy farms and AMR in human isolates from temporal and spatial matched bacterial isolates from Canada and other countries [29–31]. Recognizing that ecologically and over time, all AMU, even when appropriate, eventually contributes to AMR development and direct or indirect dissemination [32] is a key component of AMS [33]. The introduction of an AMR and One Health course within veterinary medicine curricula could be helpful to increase AMR knowledge and awareness of the link between AMU and AMR among veterinarians, as previously proposed for human medicine [34]. Importantly, awareness of AMR and AMS principles is necessary but not sufficient to reduce AMU on dairy farms, because it does not represent a practical solution for routine farm management decision-making [16]. We have shown here for veterinarians and elsewhere for farmers [21, 35] that numerous other factors influence AMU decisions, notably emphasis on the animal before them. Therefore, it is necessary to balance AMS interventions between awareness of broader issues and consideration of mindset and other influences on AMU that we have described.

It was evident that veterinarians in our sample experience some conflict when prescribing antimicrobials and trying to do it responsibly, in alignment with previous research from Northern Europe [15, 16, 36]. Veterinarians recognize themselves as stewards of animal welfare and animal production, with the professional and, in some cases, legal responsibility to promote the health and welfare and relieve suffering of animals and to address the animal care needs of the client [37], and at the same time as stewards of antimicrobials and responsible to safeguard the health and well-being of the public. This creates an ambivalent relationship with AMS [25]. On one hand veterinarians feel the pressure to offer a solution to their clients, an immediate improvement of health, to recover animal production, and to preserve animal welfare. On the other hand, livestock veterinarians are at the center of discussions regarding AMU in food-producing animals and its consequences for AMR in human and animal medicine. In addition, veterinarians from this and other studies [38] feel that they are unfairly blamed for AMR in people. This situation leads veterinarians to feel frustrated and stigmatized, which may contribute to redirection to other antimicrobial prescribers and users, such as farmers, small-animal practitioners, and human doctors and patients for inappropriate AMU and AMR. Dairy farmers from the same region manifested similar other-blaming attitudes in our previous qualitative study [21] and the same phenomenon has been reported in poultry veterinarians and human medicine [39, 40]. In our work, dairy farmers and veterinarians expressed the perception that they did not use more antimicrobials than necessary to meet their wish and duty to support animal health and welfare. If that were always accurate, any problem with AMU must lie elsewhere. However, blaming others makes it more difficult to critically reflect on room for improvement in their own antimicrobial prescribing [25]. Therefore, it is necessary to find strategies to overcome other-blaming and encourage collaborative work between farmers and veterinarians, and between veterinary and human medicine to fight AMR. We encourage comprehensive evaluations that attempt to quantify attributable fractions of contributions to AMR by AMU in dairy cattle such as reported by Tang et al. [4] to inform the urgency of behavior change by dairy farmers and veterinarians.

Some strategies and barriers to reduce AMU on dairy farms expressed by veterinarians in this study were similar to those reported by dairy farmers from the same region in our previous study [25]. Survey respondents considered that measuring and benchmarking AMU among dairy farms is the most effective strategy to reduce AMU, consistent with farmers in a survey and focus groups [21, 35]. Pragmatically, benchmarking of AMU by farmers and veterinarians may raise awareness and increase receptivity to change. According to our unpublished results of a pilot intervention strategy to reduce AMU on Canadian dairy farms, benchmarking seems to be more effective among above-average users, probably because it appeals to farmers' social desirability [41]. This was part of the "RESET model" used with legislated requirements to reduce AMU in the Netherlands [41]. The effect of benchmarking veterinary antimicrobial prescriptions among practitioners on farm-level antimicrobial stewardship has not been studied. However, benchmarking prescription patterns of veterinarians would allow veterinarians to compare their prescription practices with peers, which could be useful to identify differences in and influences on prescribing habits among veterinarians [42]. It might also increase awareness or motivate change among above-average prescribers. However, the benchmarking strategy was not mentioned during focus groups when veterinarians were asked about their ideas to reduce AMU. Instead, focus group participants centered their discussion of AMU reduction on the appropriate diagnosis of diseases and improving biosecurity and management to reduce the incidence of infectious diseases. Veterinarians are the main resource for health management, indicating that there is scope to increase their engagement in preventive medicine to improve animal health and reduce AMU on dairy farms. Improved dairy farm biosecurity was associated with reduced AMU in a study in the United States [43]. Similarly,

management strategies directed to reduce the incidence of reproductive, calf, and udder diseases reduced AMU on dairy farms in Switzerland [44]. Both veterinarians and farmers identified reviewing protocols for common diseases as a way to improve current AMU [21, 35]. This strategy facilitates good communication and trust-building between veterinarians and farmers. One experimental study found that veterinarians who feel confident in their relationship with their clients were more likely to engage in responsible antimicrobial prescribing and more likely to challenge their clients' pressure for antimicrobial prescribing [16]. Collaborative approaches between veterinarians and their clients have been shown to be effective at reducing the use of critically important antimicrobials while maintaining dairy cattle health and welfare [45]. We encourage the incorporation of this collaborative approach in the development of AMS programs in the Canadian dairy industry.

We also identified barriers to improved AMS shared between dairy cattle veterinarians and farmers. One of these was the unavailability of timely diagnostic laboratory services for microbiological culture and susceptibility tests. More than a third of the survey respondents indicated not having performed a microbiological culture and susceptibility analysis in the last year. This can be a barrier for AMS, and it has been reported in previous studies from Australia, India, the UK, and Canada [25, 46, 47]. Additional training of veterinarians and farmers regarding on-farm or in-clinic culture for clinical mastitis treatment decisions [48] and development of rapid tests of common diseases (e.g., viral vs. bacterial respiratory disease, or viral or parasitic vs. bacterial diarrhea in calves) should be part of an AMS strategy. Although veterinarians from the UK recognized the potential of rapid and point-of-care diagnostic tests in contributing to reduced AMU, they also expressed concerns about its accuracy and questioned the impact of using these tests on the flow of diagnostic and disease surveillance if it is performed by many different actors [49]. Additional research on accurate and rapid tests and the optimal way to apply these may refine and reduce AMU. Progress can also be made by implementing existing best practices based on simple diagnostic criteria and decision supports [50].

In this study, dairy cattle veterinarians manifested frustration that farmers would attempt different antimicrobial therapies before veterinary consultation and considered this practice problematic. Our previous research [35] indicates that farmers value their ability to take good care of their animals and to rely on their experience and observations, which may explain their inclination to treat cattle before contacting their veterinarian. In addition, dairy farmers reported that difficulty in prompt communication, not cost, was the main barrier to contacting the veterinarian more frequently [35]. Because consultation for every case of mild or moderate illness is not always feasible due to veterinarians' availability, ease or timeliness of communication, and current compensation models for veterinary services, it is necessary to establish a collaborative approach between veterinarians and farmers to develop usable protocols to improve AMS on Canadian dairy farms. Importantly, dairy farmers considered that it is sometimes difficult to communicate with their veterinarians due to technical language or unclear or inconsistent messaging [21, 35]. A Canadian study on dairy farmers' satisfaction with veterinary advisors found a negative association between the ratio of talk (i.e., total veterinary talk compared with total farmer talk) and farmer satisfaction, indicating an opportunity to improve communication with farmers. It is important to recognize this barrier and to establish good communication between veterinarians and farmers to optimize AMU. Training communication with farmers is possible and must be part of antimicrobial stewardship strategies. Veterinarians showed more empathic and evocative communication style, with better abilities to foster collaboration and power-sharing and more explicit verbal efforts to seize farmers' perspectives, experiences and emotions after five hours of communication training based on motivational interviewing in the UK [51].

Less than 10% of survey respondents and none of the focus group participants considered that implementing regulations would be an effective way to reduce AMU on Canadian dairy farms. In the province of Quebec, the use of category one antimicrobials (i.e., 3rd and 4th generation cephalosporins, fluoroquinolones, and polymyxin) in food-producing animals, has been restricted to clinical cases that are not treatable with an antimicrobial of a lower importance category [52]. A recent study reported a significant decrease in sales of category one antimicrobials following the implementation of the regulation [53], indicating that it is an effective measure to reduce AMU. However, this situation has created significant challenges for producers and veterinarians [54]. The lack of access to intramammary antimicrobials other than category one is an important barrier to reducing further AMU in this category [54]. Even though the number of marketed antimicrobial products is external to the government, it illustrates a dissonance between what the regulation intends and the real-life situation for the dairy industry. Additional measures to support access to treatment options in food-producing animals other than category one antimicrobials is a key action to improve AMS in Canada.

## 5.1 Limitations

We acknowledge that our survey had a relatively low response rate (16%). This is a frequent phenomenon when surveys are distributed electronically. Recent online veterinary surveys have had limited response rates [55, 56], which has been related to a general decrease in voluntarism in research due to lack of time, digital burnout, and the inability to contact participants directly due to ethics board requirements and data privacy [57]. However, responses from focus groups bear out most of the survey outcomes, which supports the survey responses. In addition, as with most surveys and focus groups, this study is subject to social desirability and self-selection bias. Nonetheless, many focus group participants shared some practices they said they were not proud of, indicating that veterinarians were willing to give honest responses. Qualitative studies are not intended to generalize results. Instead, they provide more in-depth insights into understanding, perspectives, and social meanings of a specific topic [43]. We consider that results from this study could inform the development of AMS programs, taking into account the complex role of veterinarians in responsible AMU.

We conclude that the role and positionality of veterinarians are highly influential in antimicrobial prescribing decisions. It is likely that targeted interventions for different demographic groups of veterinarians are needed to improve AMS, rather than a single and general program due to different states of awareness and risk aversion by gender, age, years of experience, and role in the veterinary clinic. We consider that there is opportunity to improve AMS on Canadian dairy farms by improving veterinarians' awareness of mechanisms of development and transmission of AMR beyond the animal and farm treated with antimicrobials, collaborative review of disease treatment protocols with dairy farmers, and additional involvement of veterinarians in preventive medicine on farms.

## Supporting information

**S1 Appendix. Questions included in the survey.**
(DOCX)

**S2 Appendix. Interview guide.**
(DOCX)

**S1 Table. Detailed demographic information of the 26 participants of the focus groups.**
(DOCX)

**S1 File.**
(DOCX)

## Author Contributions

**Conceptualization:** Claudia Cobo-Angel, Stephen J. LeBlanc.

**Data curation:** Claudia Cobo-Angel.

**Formal analysis:** Claudia Cobo-Angel, Steven M. Roche, Stephen J. LeBlanc.

**Funding acquisition:** Stephen J. LeBlanc.

**Investigation:** Claudia Cobo-Angel, Steven M. Roche.

**Methodology:** Claudia Cobo-Angel, Stephen J. LeBlanc.

**Project administration:** Stephen J. LeBlanc.

**Supervision:** Stephen J. LeBlanc.

**Validation:** Steven M. Roche.

**Visualization:** Claudia Cobo-Angel.

**Writing – original draft:** Claudia Cobo-Angel.

**Writing – review & editing:** Claudia Cobo-Angel, Steven M. Roche, Stephen J. LeBlanc.

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
