## [Decision Letter · Decision Letter 0]

6 Apr 2023

PONE-D-22-32223Understanding the role of veterinarians in antimicrobial stewardship on Canadian dairy farms: a mixed-methods studyPLOS ONE

Dear Dr. Claudia Cobo-Angel

Thank you for submitting your manuscript to PLOS ONE. First, my apologies in the delays in getting a decision to you. It has been difficult to find two reviewers to comment on the MS, for reasons which are unclear to me.  After careful consideration, we feel that it has merit but does not fully meet PLOS ONE’s publication criteria as it currently stands. Therefore, we invite you to submit a revised version of the manuscript that addresses the points raised during the review process.  Both reviewers highlight the importance of the research topic and relevance of the mixed methods approach adopted but recommend some restructuring of the manuscript.  The first reviewer recommends major revision and provides a list of specific issues to be addressed. Importantly, the reviewer raises issues regarding the thematic analysis and interpretation of results, which need to be addressed in the resubmission process. The second reviewer recommends minor revisions, Reviewer 2 again provides a list of specific comments to be addressed, but also indicates the need for some restructuring of the results: specifically, integration of the quantitative and qualitative results rather than separately presented. In addition, a sample size analysis for the survey needs to be included. reviewer comments are copied below. 

Please submit your revised manuscript by May 21 2023 11:59PM If you will need more time than this to complete your revisions, please reply to this message or contact the journal office at plosone@plos.org. Please include the following items when submitting your revised manuscript:A rebuttal letter that responds to each point raised by the academic editor and reviewer(s). You should upload this letter as a separate file labeled 'Response to Reviewers'.A marked-up copy of your manuscript that highlights changes made to the original version. You should upload this as a separate file labeled 'Revised Manuscript with Track Changes'.An unmarked version of your revised paper without tracked changes. You should upload this as a separate file labeled 'Manuscript'.

We look forward to receiving your revised manuscript.

Kind regards,

Lynn Jayne Frewer, MSc PhD

Academic Editor

PLOS ONE

Journal Requirements:

Reviewers' comments:

Reviewer's Responses to Questions

**Comments to the Author**

1. Is the manuscript technically sound, and do the data support the conclusions?

Reviewer #1: Yes

Reviewer #2: Yes

2. Has the statistical analysis been performed appropriately and rigorously? 

Reviewer #1: Yes

Reviewer #2: Yes

3. Have the authors made all data underlying the findings in their manuscript fully available?

Reviewer #1: Yes

Reviewer #2: No

4. Is the manuscript presented in an intelligible fashion and written in standard English?

Reviewer #1: Yes

Reviewer #2: Yes

5. Review Comments to the Author

Reviewer #1: Overall comment

Excellently written and an important subject to understand better. Good choice to use mixed methods to better understand a complex social process of decision-making, negotiation, identity and professional culture.

I have some concerns about the ‘understanding the role of veterinarians’ part as this is a very heavily researched area and not sure how well this study, as it stands, adds to our knowledge. Although I accept this is in the context of Canada and dairy farming, which does add novelty to some extent.

Introduction and methods

Good opening paragraph

Line 57 – However makes vets sound like the exception, in what context? I would say ‘In addition, veterinarians are commonly central to AMS on farms in many parts of the world.’

Line 67 – below are some helpful papers to include about why there’s a need to understand a vet’s role in AMS

https://bvajournals.onlinelibrary.wiley.com/doi/full/10.1136/vr.105041

https://www.journalofdairyscience.org/article/S0022-0302(21)00141-7/fulltext

Line 70 – OIE is now WOAH

Line 81 – the word sound is a little subjective. Consider use of word responsible/prudent/better…

Line 86 – A reference or example would be good here

Line 112 – ‘their right to choose not to answer…’

Line 149 – what is listserv?

Positionality and reflexivity statement

Line 166 – excellent!

Line 173-175 – doesn’t quite make sense.

Line 176 – Perhaps, rephrase… ‘differences in sociocultural background between CC-A and the participants arises due to the fact CC-A is not Canadian…’

Consider including your research paradigm (positivism>>interpretivism?)

Results

Line 193-4 – Interesting fact to start. Quite high I think!

Line 197 – The differences in pronouns amongst participants in surveys v focus groups and the preference for she/her to cite peer reviewed literature is intriguing. Might be justification to talk about your data through a feminist lens?

Where is Figure 1 and 2 as not in line with the text?

Line 212-214 – The fact more he/him’s rated farmers’ goals higher is another interesting finding. To discuss!

Line 218 – Instead of ‘Most’ I would use ‘majority’ as more precise language

Line 224- this is surprising! My experience is younger vets are more inclined to micro test than more experienced vets.

Line 235 - extra ‘the’

Lines 247-251 – again another interesting result re difference between participant pronouns! Worth discussing, see later points.

Lines 274-281 – This is an interesting finding from the survey – most participants felt their clients were using AM responsibly and alongside guidance. I wonder how this contrasts with FG data!

Line 292 – I would not say ‘among others’, I would list them all if summarising the main themes. You also said 3 main themes and then list 4 things, which is a bit unclear then what you are summarising.

No figure 3 in line with text

Line 334-335 – might be worth defining what you mean by ‘extra label’ use earlier in paper for non-vets/international audiences as can be described differently in different places

Lines 351-352 – Love this quote.

Lines 355-357 – This is a shock! So, practice owners admitted they will choose certain drugs over others because they have a good relationship with a drug company? Is this permitted in Canada? What are the ethical and legal implications?

Lines 419-422 – An interesting point made! Why are many drugs being removed from the veterinary market? How does this process work in Canada and what influences it? Potential things for discussion.

Line 437 – would be good to have a contrasting quote here.

Line 521 – impediments to ‘using’, maybe re word sentence to make it explicit it is the long distance to deliver samples and long result wait time

Line 537-541 – Very interesting. This speaks to the vet’s proficiency in communication and knowledge exchange. What else did participants say about it and what does this mean for farmer-vet comms, relationships and improved AMS?

Line 542-546 – Your statement and quote do not really align. Awareness of AMR, I would argue, from your results is high, but an appreciation of the relevance and implications is limited, which is what P25_4 shows.

Discussion

Line 557 – I would choose to use the word personal or individual assuming you mean same thing. If not define what you mean. Same goes for external or contextual.

Line 569 – missing ]

Line 575 – irregular reference style

Line 585 – I would qualify this statement so it is clear what you mean ‘problem in dairy cattle’. So problem for cattle health from not being able to treat resistant infections or ‘problem in dairy cattle’ as any excessive use is contributing to resistant infections that can then spread in the environment, to other animals and people?

Line 592-600 – I would discuss the different approaches to risk within AMS here, some people more risk averse than others. There is a risk of AMR but some participants are not convinced the risk is high or that what they do increases the risk… What influences these risk-based behaviours? What is the implication of risk-aversion in light of your results? You mention the risk of any AMU leading to AMR, you could link these together a bit more and in more depth.

LINE 600 – See this paper for evidence/context https://research-information.bris.ac.uk/en/publications/limited-phylogenetic-overlap-between-fluoroquinolone-resistant-es

Line 607 – good!

Line 617 – Could you add some detail and insight into the contextual restraints that govern and influence veterinarians and therefore, their decision making. They feel a sense of stewardship for animal welfare but it is more than just a moral sense, veterinarians are professionally bound and, in some cases, legally bound to act in the interests of animal welfare.

Line 622- do you have quote to illustrate this or reference?

Line 643-644 – I would reference why you claim benchmarking is more likely to affect above average users and not affect all users to some degree.

Line 649 - Why? This needs much more explaining if you are to include it. Why would some farmers be more motivated by benchmarking than some vets? Are we simply lacking in data of the efficacy of benchmarking vets re AMS? What if a vet is also a farmer? What do these labels do to our perception of how motivated by benchmarking an individual might be?

Line 654-656 – There are quite a few studies that have demonstrated this, why pick Switzerland? Might strengthen your point to include 3 different areas/studies, including in Canada or N.America.

Line 677 – I would read and include the following reference here https://www.frontiersin.org/articles/10.3389/fvets.2020.569545/full

Not just a case of including more accurate rapid tests – the role of the vet and actions of vets is key to maximising success of diagnostics in AMS, which ties in nicely with a key theme in your results – role of vets and their positions in relation to farms and the wider profession.

Line 697 – not clear what you refer above to?

Lines 691-701 – This very superficially skims over a huge element of AMS – communication and helping people change. I would consider discussing how the vet communicates (lots in literature about this) and how vets build relationships with farmer clients and therefore, how this affects the vet’s ability to practice AMS effectively i.e. reduce AMU.

Line 702 – I would consider a subheading called Limitations

Line 703-707 – good explanatory description here

Line 707 – consider phrasing. I would say ‘support survey responses’

Line 713 – I would add ‘more in-depth insights’ and rephrase to say ‘more in-depth insights into perspectives, behaviours and shared meanings on a specific topic, which helps develop our understanding of social phenomena, such as prescribing practices in the veterinary context.’

Line 718 – Your results demonstrate that vet awareness of AMR is high, they just feel dissonance about it so this line is inconsistent with your findings. The other 2 points “collaborative review of disease treatment protocols with dairy farmers, and additional involvement of veterinarians in preventive medicine on farms” are valid. I would argue that the really big take home message from your results is the role and positionality of the vet is critical in how they practice AMS and their prescribing decisions. Your results illustrate the vets’ gender/pronouns, age, experience, role in practice and position relative to other vets has a major influence on their perspectives and self-reported prescribing behaviour - in the survey and focus groups – can you pull out any more data on this? This key finding now needs further interpretation and discussion based on other literature and then embedded in your results.

Found the figures! Would be better in line with text and the quality is not good so hard to see the text labels.

Comment on themes/overall findings

I always link my themes from my qualitative research back to my research questions and what questions/gaps the research has answered. I would recommend reviewing your theme titles too as they do not directly answer your questions or speak to your paper title. Your themes need to help us understand the role of the veterinarian in AMS on Canadian dairy farms. For instance, theme 3 is simply descriptive and not interpretative. Yes, there are opportunities and barriers to AMS but what do we need to understand about the differing attitudes of vets, the slow evolution of AMS/change, the need to work collaboratively and make use of things like benchmarking and enhanced relationships between farmers -vets…? Think when you write about your results…Ok, so what? What does this mean?

Based on what you wrote at the beginning of the paper…

‘describe the perceptions and influences in AMU decision-making by dairy veterinarians.’

This is described well in this paper and your methods are robust and well conveyed. The tricky bit is then interpreting what that means!

Considering the complexity and multifactorial nature of veterinarians’ prescribing decisions, we conducted mixed methods research to specifically explore dairy cattle veterinarians’ considerations for antimicrobial prescribing,

Mixed methods is an appropriate choice for your research questions and adds to the validity of your findings. The research has identified specific considerations for AM prescribing, which you detail well (farm and animal specifics, availability of drugs, animal welfare, farmer compliance, other vets…). This is reflected in theme 1 and 2 – veterinary knowledge, age experience and role in clinic, conflict between animal welfare and AMS, farm specifics, other vets, drug availability and drug company influence. However, it is not clear as to the difference between the 2 themes and what the reader needs to understand about the vet’s role in AMS.

I would be inclined to refine ‘veterinary knowledge’ to be specific about ‘veterinary knowledge of drug labels and treatment regimes’.

I would encourage you to delve deeper into the implications for AMS in Canada based on the veterinarian’s role, see earlier comments. I think you could say more about the intersection between the demographics of your participants (age and gender), their positionality in the profession, with farmers and with one another and what this means for AMS.

This then ties in nicely to the 3rd sub theme, which is the tension between maintaining welfare and AMS, which you excellently describe.

In light of all that, I would encourage you to slightly rethink the name of theme 1. Your paper is titled ‘understanding the role of veterinarians in AMS…’, so what does theme 1 tell us that improves our knowledge/understanding of the vet’s role? That decision making depends on vet education (not sure your data mentions this) or knowledge of available drugs and treatment regimens? That decision making is dependent on the complex interaction between a veterinarian’s (perceived) role (and responsibilities to animal welfare?), experience of AM, the veterinarian’s wider status in the profession/with farmers? This then starts to overlap with theme 2. Perhaps theme 1 + 2 are speaking about the same thing but you have attempted to split internal factors and external ones? If this is what you want, then it needs to be clearer.

I would suggest Theme 1 could be the ‘Interconnected nature of the veterinarian as a person, professional and scientist’ which could have the subthemes - gender and experience, role within the profession and on farm, conflict between animal welfare and AMS, drug knowledge and availability. Theme 2 could then be ‘Veterinarians as critical partners in AMS’ with subthemes – uncertainty about dairy farming’s role in AMR, AMS as an evolving topic, barriers to AMS, ways to reduce usage. Just ideas from what you’ve presented.

their attitudes toward reducing AMU, and awareness of AMR,

As highlighted earlier, be careful not to conflate awareness (or lack of it) with uncertainty about links between AMR-AMU in farming / influence of AMU in dairy cattle on AMR in humans. I think comparing how there is high awareness of AMR in your study back to FAO/WHO guidelines is very good. You also capture differing attitudes to AMR and how that relates to the veterinarian’s experience/role/status, which is documented elsewhere in the literature. What does this mean for improving AMS in Canada? Who and what needs to be accounted for in any intervention/initiative based on your findings?

perceived barriers to improving AMS on dairy farms in Canada

These were documented succinctly and solutions/strategies to overcome them proposed, including by participants, which is almost getting into action research territory and highlights the benefits of collaborative initiatives to improve AMS. It might be worth making this explicit if that was indeed the case, that your focus group participants came up with ways to improve AMS in Canada. Bottom up!

Comments on Interview guide

- There are some quite leading and loaded questions, which could bias your results/limit finding out other influencing factors e.g.

Q6. How do farmers expectations of how they think the animal should be treated influence your decision-making around antimicrobial use?”

Q7. How does the cost of treatment influence decision-making?

Q7. How do your colleagues affect your antimicrobial prescribing decisions?

Probe: do you discuss antimicrobial treatment decisions with your colleagues?

Q8. How do pharmaceutical companies affect your antimicrobial prescribing decisions?

Q9. How does the current public discussion on AMU/AMS influence your decisions?

- I would have asked ‘Tell me about a time when you have prescribed AM?’ ‘What do you consider before prescribing?’ ‘What might affect your decisions to prescribe or not, or to use an AM or not?’

- Just as a word of warning, the questions you ask in the interviews/focus groups should not mirror your qualitative analysis themes as this suggests limited data analysis and interpretation, which is the whole purpose of thematic analysis. Otherwise they could be asked in a survey and described and analysed descriptively.

Reviewer #2: This is a well presented manuscript describing a mixed method study of barriers to AMS in dairy veterinarians.

My main suggestion is that I would prefer the quantitative and qualitative results be integrated rather than presented separately - this is more keeping with a mixed-method approach and allows for contrasting and highlighting consistencies. It may also shorten the manuscript, which is quite long. You could also consider integrating the discussion with the results.

Minor comments:

Method - did you do a sample size analysis for the survey? This should be included in the method, even if it is post-hoc.

Line 173 - Sentence should read "As SUCH veterinarian CC-A"

Through-out quantitative results - please include the magnitude of the difference/change or the proportions for each group rather than just the P-value.

Line 569: Reference needs a closing bracket

Line 575: Hockenhull reference needs reformatting. You could also consider Scarborough et al, 2023 - Brave Enough in this part of the discussion.

Line 603-605: I think there is ample AMR and OH in the veterinary medicine curricula, and the authors point out that new grads are more likely to be aware of AMR. Recommend omitting this sentence.

Figures - quality isn't great - need higher resolution images.

Figure 3 doesn't add much to the text.

6. PLOS authors have the option to publish the peer review history of their article (what does this mean?). If published, this will include your full peer review and any attached files.

Reviewer #1: No

Reviewer #2: No

---

## [Author Response · Author response to Decision Letter 0]

7 Jul 2023

Dear editor and reviewers,

Thank you for your work and suggestions on our manuscript. We consider that the manuscript has substantially improved after including your changes and suggestions. 

REVIEWERS’ COMMENTS

Reviewer #1: Overall comment

Excellently written and an important subject to understand better. Good choice to use mixed methods to better understand a complex social process of decision-making, negotiation, identity, and professional culture.

I have some concerns about the ‘understanding the role of veterinarians’ part as this is a very heavily researched area and not sure how well this study, as it stands, adds to our knowledge. Although I accept this is in the context of Canada and dairy farming, which does add novelty to some extent.

Authors: Thank you for your comments and input on our work. We consider that there is room for research on the veterinarians’ role in antimicrobial stewardship, particularly in the local context. Improving antimicrobial stewardship involves several considerations, not only at the clinical level. As we could see in our results, veterinarians had numerous and sometimes contradictory thoughts regarding antimicrobial prescribing beyond animal diagnosis, involving regulatory, social, and economic factors that need to be understood and considered to develop and implement antimicrobial stewardship programs. These findings help to advance understanding of the subject and contribute to informing the development of policy in this area. We also note that PlosOne emphasizes validity rather than novelty. 

Introduction and methods

Good opening paragraph

Line 57 – However makes vets sound like the exception, in what context? I would say ‘In addition, veterinarians are commonly central to AMS on farms in many parts of the world.’

Authors: wording corrected

Line 67 – below are some helpful papers to include about why there’s a need to understand a vet’s role in AMS

https://bvajournals.onlinelibrary.wiley.com/doi/full/10.1136/vr.105041

https://www.journalofdairyscience.org/article/S0022-0302(21)00141-7/fulltext

Line 70 – OIE is now WOAH

Authors: corrected

Line 81 – the word sound is a little subjective. Consider use of word responsible/prudent/better…

Authors: wording corrected

Line 86 – A reference or example would be good here

Authors: examples and references were included 

Line 112 – ‘their right to choose not to answer…’

Authors: wording corrected 

Line 149 – what is listserv?

Authors: It is a software to manage email lists. But there is no need to mention it and was removed

Positionality and reflexivity statement

Line 166 – excellent!

Line 173-175 – doesn’t quite make sense.Line 176 – Perhaps, rephrase… ‘differences in sociocultural background between CC-A and the participants arises due to the fact CC-A is not Canadian…’

Consider including your research paradigm (positivism>>interpretivism?)

Authors: rephrased to improve clarity

Results

Line 193-4 – Interesting fact to start. Quite high I think!

Line 197 – The differences in pronouns amongst participants in surveys v focus groups and the preference for she/her to cite peer reviewed literature is intriguing. Might be justification to talk about your data through a feminist lens?

 Authors: we included a discussion regarding pronoun identity and reported prescribing behavior. We appreciate the suggestion, but we prefer to maintain the current scope of the analysis, in which we tried to reflect the focus group discussions rather than superimpose any particular perspective. 

Where is Figure 1 and 2 as not in line with the text?

Authors: PlosOne requires figures to be at the end of the text. However, these figures were moved to supplementary material. 

Line 212-214 – The fact more he/him’s rated farmers’ goals higher is another interesting finding. To discuss!

Authors: we included a discussion regarding pronoun identity and reported prescribing behavior (see lines 380-389)

Line 218 – Instead of ‘Most’ I would use ‘majority’ as more precise language

Authors: wording corrected

Line 224- this is surprising! My experience is younger vets are more inclined to micro test than more experienced vets.

Line 235 - extra ‘the’

Authors: corrected

Lines 247-251 – again another interesting result re difference between participant pronouns! Worth discussing, see later points.

Authors: we included a discussion regarding pronoun identity and reported prescribing behavior (see lines 380-389)

Lines 274-281 – This is an interesting finding from the survey – most participants felt their clients were using AM responsibly and alongside guidance. I wonder how this contrasts with FG data!

Line 292 – I would not say ‘among others’, I would list them all if summarizing the main themes. You also said 3 main themes and then list 4 things, which is a bit unclear then what you are summarizing.

Authors: rephrased to improve clarity

Line 334-335 – might be worth defining what you mean by ‘extra label’ use earlier in paper for non-vets/international audiences as can be described differently in different places

Authors: an extra-label antimicrobial use definition was added 

Lines 351-352 – Love this quote.

Lines 355-357 – This is a shock! So, practice owners admitted they will choose certain drugs over others because they have a good relationship with a drug company? Is this permitted in Canada? What are the ethical and legal implications?

Authors: veterinarians have the freedom to select which antimicrobial products they prescribe. We rephrased and added an additional quote to clarify that this is related to the product brand, not to the choice of antimicrobial prescribed (e.g., brand A vs. brand B of penicillin) (Table 2).

Lines 419-422 – An interesting point was made! Why are many drugs being removed from the veterinary market? How does this process work in Canada and what influences it? Potential things for discussion.

Authors: Two intramammary products are not available anymore, which is not related to AMS regulations, but pharmaceutical companies’ business decisions or supply issues. We added discussion at this regard (see line 559). 

Line 437 – would be good to have a contrasting quote here.

Authors: A quote was included (Table 4. Under subtheme 3a Attitudes toward AMR and AMS)

Line 521 – impediments to ‘using’, maybe re word sentence to make it explicit it is the long distance to deliver samples and long result wait time.

Authors: rephrased for clarity 

Line 537-541 – Very interesting. This speaks to the vet’s proficiency in communication and knowledge exchange. What else did participants say about it and what does this mean for farmer-vet comms, relationships and improved AMS?

Authors: We discussed the veterinarian communication and its role in AMS (lines 537-550)

Line 542-546 – Your statement and quote do not really align. Awareness of AMR, I would argue, from your results is high, but an appreciation of the relevance and implications is limited, which is what P25_4 shows.

Authors: We rephrased for clarity 

Discussion

Line 557 – I would choose to use the word personal or individual assuming you mean same thing. If not define what you mean. Same goes for external or contextual.

Authors: wording corrected

Line 569 – missing ]

Authors: corrected

Line 575 – irregular reference style

Authors: corrected

Line 585 – I would qualify this statement so it is clear what you mean ‘problem in dairy cattle’. So problem for cattle health from not being able to treat resistant infections or ‘problem in dairy cattle’ as any excessive use is contributing to resistant infections that can then spread in the environment, to other animals and people?

Authors: Good point, we did not specify what kind of problem for dairy cattle. Our intent was to refer to visible health problems in participants’ animals. Participants could have interpreted this in different ways, and still, less than half of the respondents didn’t consider it a problem. This was added to the discussion (lines 412 – 416).

Line 592-600 – I would discuss the different approaches to risk within AMS here, some people more risk averse than others. There is a risk of AMR but some participants are not convinced the risk is high or that what they do increases the risk… What influences these risk-based behaviours? What is the implication of risk-aversion in light of your results? You mention the risk of any AMU leading to AMR, you could link these together a bit more and in more depth.

Authors: We included the risk perception concept in the discussion and how it is implied in AMS and prescribing behavior (lines 412 – 437)

LINE 600 – See this paper for evidence/context https://research-information.bris.ac.uk/en/publications/limited-phylogenetic-overlap-between-fluoroquinolone-resistant-es

Authors: We included this and other papers in our discussion (lines 432 – 434)

Line 607 – good!

Line 617 – Could you add some detail and insight into the contextual restraints that govern and influence veterinarians and therefore, their decision making. They feel a sense of stewardship for animal welfare but it is more than just a moral sense, veterinarians are professionally bound and, in some cases, legally bound to act in the interests of animal welfare.

Authors: that’s correct. We added it to our discussion (lines 450 – 454)

30. Line 622- do you have quote to illustrate this or reference?

Authors: we quoted this subtheme 3a “Attitudes toward AMR and AMS”. Additionally, we included an additional reference to poultry veterinarians and human medicine (line 465) 

Line 643-644 – I would reference why you claim benchmarking is more likely to affect above average users and not affect all users to some degree.

Authors: we clarified rephrasing to: “According to our unpublished results of an intervention strategy to reduce AMU on Canadian dairy farms, benchmarking seems to be more effective among above-average users, probably because it appeals to farmers’ social desirability [41]” (lines 482 – 485)

Line 649 - Why? This needs much more explaining if you are to include it. Why would some farmers be more motivated by benchmarking than some vets? Are we simply lacking in data of the efficacy of benchmarking vets re AMS? What if a vet is also a farmer? What do these labels do to our perception of how motivated by benchmarking an individual might be?

Authors: we agree that there is a lack of studies regarding the utility or perceptions of benchmarking prescriptions among veterinarians. We rephrased it as follows:

 “The effect of benchmarking veterinary antimicrobial prescriptions among practitioners on farm-level antimicrobial stewardship has not been studied. However, benchmarking prescription patterns of veterinarians would allow veterinarians to compare their prescription practices with peers, which could be useful to identify differences in and influences on prescribing habits among veterinarians [42]. It might also increase awareness or motivate change among above-average prescribers.”

Line 654-656 – There are quite a few studies that have demonstrated this, why pick Switzerland? Might strengthen your point to include 3 different areas/studies, including in Canada or N.America.

Authors: A reference from North America was included (line 499)

Line 677 – I would read and include the following reference here https://www.frontiersin.org/articles/10.3389/fvets.2020.569545/full

Not just a case of including more accurate rapid tests – the role of the vet and actions of vets is key to maximising success of diagnostics in AMS, which ties in nicely with a key theme in your results – role of vets and their positions in relation to farms and the wider profession.

Authors: Thank you for the suggestion, we included the following information (lines 541 – 527): 

“Although veterinarians from the UK recognized the potential of rapid and point-of-care diagnostic tests in contributing to reduce AMU, they also expressed concerns about its accuracy and questioned the impact of using these tests on the flow of diagnostic and disease surveillance if it is performed by many different actors [49]. Additional research on accurate and rapid tests and the optimal way to apply these may refine and reduce AMU”.

Lines 691-701 – This very superficially skims over a huge element of AMS – communication and helping people change. I would consider discussing how the vet communicates (lots in literature about this) and how vets build relationships with farmer clients and therefore, how this affects the vet’s ability to practice AMS effectively i.e. reduce AMU.

Authors: We expanded the discussion regarding communication between veterinarians and farmers and the implications on AMS (lines 542 – 545)

Line 702 – I would consider a subheading called Limitations

Authors: Included

Line 703-707 – good explanatory description here

Line 707 – consider phrasing. I would say ‘support survey responses’

Authors: Rephrased 

Line 713 – I would add ‘more in-depth insights’ and rephrase to say ‘more in-depth insights into perspectives, behaviours and shared meanings on a specific topic, which helps develop our understanding of social phenomena, such as prescribing practices in the veterinary context.’

Authors: Rephrased 

Line 718 – Your results demonstrate that vet awareness of AMR is high, they just feel dissonance about it so this line is inconsistent with your findings. The other 2 points “collaborative review of disease treatment protocols with dairy farmers, and additional involvement of veterinarians in preventive medicine on farms” are valid. I would argue that the really big take home message from your results is the role and positionality of the vet is critical in how they practice AMS and their prescribing decisions. Your results illustrate the vets’ gender/pronouns, age, experience, role in practice and position relative to other vets has a major influence on their perspectives and self-reported prescribing behaviour - in the survey and focus groups – can you pull out any more data on this? This key finding now needs further interpretation and discussion based on other literature and then embedded in your results.

Authors: The conclusion paragraph was rephrased (lines 585 – 594)

Found the figures! Would be better in line with text and the quality is not good so hard to see the text labels.

Authors: The figures were moved to supplementary material

Comment on themes/overall findings

I always link my themes from my qualitative research back to my research questions and what questions/gaps the research has answered. I would recommend reviewing your theme titles too as they do not directly answer your questions or speak to your paper title. Your themes need to help us understand the role of the veterinarian in AMS on Canadian dairy farms. For instance, theme 3 is simply descriptive and not interpretative. Yes, there are opportunities and barriers to AMS but what do we need to understand about the differing attitudes of vets, the slow evolution of AMS/change, the need to work collaboratively and make use of things like benchmarking and enhanced relationships between farmers -vets…? Think when you write about your results…Ok, so what? What does this mean?

Based on what you wrote at the beginning of the paper…

‘describe the perceptions and influences in AMU decision-making by dairy veterinarians.’ This is described well in this paper and your methods are robust and well conveyed. The tricky bit is then interpreting what that means!

Considering the complexity and multifactorial nature of veterinarians’ prescribing decisions, we conducted mixed methods research to specifically explore dairy cattle veterinarians’ considerations for antimicrobial prescribing,

Mixed methods is an appropriate choice for your research questions and adds to the validity of your findings. The research has identified specific considerations for AM prescribing, which you detail well (farm and animal specifics, availability of drugs, animal welfare, farmer compliance, other vets…). This is reflected in theme 1 and 2 – veterinary knowledge, age experience and role in clinic, conflict between animal welfare and AMS, farm specifics, other vets, drug availability and drug company influence. However, it is not clear as to the difference between the 2 themes and what the reader needs to understand about the vet’s role in AMS.

I would be inclined to refine ‘veterinary knowledge’ to be specific about ‘veterinary knowledge of drug labels and treatment regimes’.

Authors: we renamed this subtheme. Thanks for the suggestion. 

I would encourage you to delve deeper into the implications for AMS in Canada based on the veterinarian’s role, see earlier comments. I think you could say more about the intersection between the demographics of your participants (age and gender), their positionality in the profession, with farmers and with one another and what this means for AMS.

This then ties in nicely to the 3rd sub theme, which is the tension between maintaining welfare and AMS, which you excellently describe.

In light of all that, I would encourage you to slightly rethink the name of theme 1. Your paper is titled ‘understanding the role of veterinarians in AMS…’, so what does theme 1 tell us that improves our knowledge/understanding of the vet’s role? That decision making depends on vet education (not sure your data mentions this) or knowledge of available drugs and treatment regimens? That decision making is dependent on the complex interaction between a veterinarian’s (perceived) role (and responsibilities to animal welfare?), experience of AM, the veterinarian’s wider status in the profession/with farmers? This then starts to overlap with theme 2. Perhaps theme 1 + 2 are speaking about the same thing but you have attempted to split internal factors and external ones? If this is what you want, then it needs to be clearer.

I would suggest Theme 1 could be the ‘Interconnected nature of the veterinarian as a person, professional and scientist’ which could have the subthemes - gender and experience, role within the profession and on farm, conflict between animal welfare and AMS, drug knowledge and availability. Theme 2 could then be ‘Veterinarians as critical partners in AMS’ with subthemes – uncertainty about dairy farming’s role in AMR, AMS as an evolving topic, barriers to AMS, ways to reduce usage. Just ideas from what you’ve presented. As highlighted earlier, be careful not to conflate awareness (or lack of it) with uncertainty about links between AMR-AMU in farming / influence of AMU in dairy cattle on AMR in humans. I think comparing how there is high awareness of AMR in your study back to FAO/WHO guidelines is very good. You also capture differing attitudes to AMR and how that relates to the veterinarian’s experience/role/status, which is documented elsewhere in the literature. What does this mean for improving AMS in Canada? Who and what needs to be accounted for in any intervention/initiative based on your findings?

perceived barriers to improving AMS on dairy farms in Canada

Authors: We appreciate the suggestions. We renamed theme 1 and expanded the interpretation and considerations of the implications of the study for AMS on Canadian dairy farms. However, we are conscious of reflecting what was provided by participants without over-interpretation or extrapolation. 

These were documented succinctly and solutions/strategies to overcome them proposed, including by participants, which is almost getting into action research territory and highlights the benefits of collaborative initiatives to improve AMS. It might be worth making this explicit if that was indeed the case, that your focus group participants came up with ways to improve AMS in Canada. Bottom up!

Comments on Interview guide

- There are some quite leading and loaded questions, which could bias your results/limit finding out other influencing factors e.g.

Q6. How do farmers expectations of how they think the animal should be treated influence your decision-making around antimicrobial use?”

Q7. How does the cost of treatment influence decision-making?

Q7. How do your colleagues affect your antimicrobial prescribing decisions?

Probe: do you discuss antimicrobial treatment decisions with your colleagues?

Q8. How do pharmaceutical companies affect your antimicrobial prescribing decisions?

Q9. How does the current public discussion on AMU/AMS influence your decisions?

- I would have asked ‘Tell me about a time when you have prescribed AM?’ ‘What do you consider before prescribing?’ ‘What might affect your decisions to prescribe or not, or to use an AM or not?’

- Just as a word of warning, the questions you ask in the interviews/focus groups should not mirror your qualitative analysis themes as this suggests limited data analysis and interpretation, which is the whole purpose of thematic analysis. Otherwise they could be asked in a survey and described and analyzed descriptively.

Authors: Thanks for your comments and suggestions on the interview guide, we will consider them for future studies. 

Reviewer #2: This is a well presented manuscript describing a mixed method study of barriers to AMS in dairy veterinarians.

My main suggestion is that I would prefer the quantitative and qualitative results be integrated rather than presented separately - this is more keeping with a mixed-method approach and allows for contrasting and highlighting consistencies. It may also shorten the manuscript, which is quite long. You could also consider integrating the discussion with the results.

Authors: Thanks for the suggestion. We integrated the quantitative and qualitative results. 

Minor comments:

Method - did you do a sample size analysis for the survey? This should be included in the method, even if it is post-hoc.

Authors: we did not calculate a sample size because the aim of this study is not to generalize results.

Line 173 - Sentence should read "As SUCH veterinarian CC-A"

Authors: the sentence was rephrased “However, as veterinarian CC-A understands the role of antimicrobial use on dairy cattle health and production and the complexity of the prescribing decisions” (line 181)

Through-out quantitative results - please include the magnitude of the difference/change or the proportions for each group rather than just the P-value.

Authors: thanks for the suggestion. The difference in proportions was included. 

Line 569: Reference needs a closing bracket

Authors: Corrected

Line 575: Hockenhull reference needs reformatting. You could also consider Scarborough et al, 2023 - Brave Enough in this part of the discussion.

Authors: Reference was corrected and Scarborough et al, 2023 was included in the discussion (line 459)

Line 603-605: I think there is ample AMR and OH in the veterinary medicine curricula, and the authors point out that new grads are more likely to be aware of AMR. Recommend omitting this sentence.

Authors: Thank you for your suggestions. We advocate inclusion of AMR in the one health context to improve knowledge and awareness of AMR transmission mechanisms, not only in Canada but other countries. 

Figures - quality isn't great - need higher resolution images.

Figure 3 doesn't add much to the text.

Authors: Figures were moved to supplementary material with improved quality

---

## [Editor Report · Decision Letter 1]

19 Jul 2023

Understanding the role of veterinarians in antimicrobial stewardship on Canadian dairy farms: a mixed-methods study

PONE-D-22-32223R1

Dear Dr. Claudia Cobo-Angel

We’re pleased to inform you that your manuscript has been judged scientifically suitable for publication and will be formally accepted for publication once it meets all outstanding technical requirements.

Kind regards,

Lynn Jayne Frewer, MSc PhD

Academic Editor

PLOS ONE

---

## [Editor Report · Acceptance letter]

20 Jul 2023

PONE-D-22-32223R1 

Understanding the role of veterinarians in antimicrobial stewardship on Canadian dairy farms: a mixed-methods study 

Dear Dr. Cobo-Angel:

I'm pleased to inform you that your manuscript has been deemed suitable for publication in PLOS ONE. Congratulations! Your manuscript is now with our production department. 

Kind regards, 

on behalf of

Dr. Lynn Jayne Frewer 

Academic Editor

PLOS ONE